# A Survey on Optical Coherence Tomography—Technology and Application

**DOI:** 10.3390/bioengineering12010065

**Published:** 2025-01-14

**Authors:** Ali Mokhtari, Bogdan Mihai Maris, Paolo Fiorini

**Affiliations:** 1Department of Computer Science, University of Verona, 37134 Verona, Italy; ali.mokhtari@univr.it; 2Department of Engineering for Innovation Medicine, University of Verona, 37134 Verona, Italy; bogdan.maris@univr.it

**Keywords:** OCT, biomedical imaging, clinical application, artificial intelligence

## Abstract

This paper reviews the main research on Optical Coherence Tomography (OCT), focusing on the progress and advancements made by researchers over the past three decades in its methods and medical imaging applications. By analyzing existing studies and developments, this review aims to provide a foundation for future research in the field.

## 1. Introduction

OCT emerged in the early 1990s as a groundbreaking imaging technique within the health automation trend. Utilizing near-infrared light, OCT enables the generation of cross-sectional images of tissues and structures without physical intrusion or contact. This capability has established OCT as an invaluable tool in the field of ophthalmology, facilitating the diagnosis and management of various ocular disorders.

With ongoing advancements, OCT is increasingly being adopted across other medical specialties, including dermatology, cardiology, and oncology. This expansion enhances our understanding of disease pathogenesis and therapeutic efficacy.

This paper explores the fundamental concepts, recent advancements, and expanding applications of OCT in clinical practice, emphasizing its growing significance in medicine. Additionally, it examines the impact of OCT on improving treatment outcomes and quality of life for patients.

OCT has been primarily used for imaging ocular tissues during its first two decades of application [1,2,3,4,5,6,7,8]. Its broad range of applications warrants further technological development in the field of eye care diagnostics. Much of the research work conducted so far involving the practical use of OCT, as presented in various papers, has focused on the imaging results of ocular tissues.

In order to demonstrate normal anatomic variations and the possible uses of OCT in practice, several types of OCT scans were performed. These include scans along the axis of the papillomacular region, series of so-called sagittal sequence macular tomograms, radial sequence tomograms of the optic disc, and circular peripapillary tomograms. These scans illustrate the facility of OCT in profiling the normal anatomical variation of the retina in terms of thickness, and the nerve fiber layer changes which are important in the diagnosis and monitoring of diseases such as glaucoma [9,10,11,12,13].

The first reason for this focus stems from the limited penetration depth of light in tissues. Typically, the depth of light transmission in tissues does not exceed one to three millimeters, making OCT alone unsuitable for imaging relatively thick tissues.

The second reason relates to the necessity for a linear scan of the tissue slice, which requires the movement of an optical fiber or waveguide along that line. Given the demand for absolute nanometric resolution, robotic arms can achieve high precision; however, they are often time consuming. In contrast, most techniques employ a movable reflective mirror, which significantly enhances scanning speed [14]. This approach, however, necessitates considerable space for the mechanical components that facilitate mirror movement. As a result, these devices must be positioned outside the patient’s body. Considering the specific conditions of the eye and the transparency of intraocular fluid, this configuration is well suited for patients undergoing ocular imaging.

In recent years, thicker tissues have become accessible for medical imaging due to the combined use of ultrasound and OCT imaging, as well as the implementation of rapid robotic systems equipped with transparent tubes that enable simultaneous imaging. [15,16].

In the realm of tumor detection, various imaging techniques complement the use of OCT. For instance, phototherapy has gained prominence due to its selectivity and minimal side effects, particularly in treating cervical cancer, for which hollow mesoporous manganese dioxide nanoparticles are utilized to enhance the delivery of photosensitizers like indocyanine green (ICG). Additionally, the application of near-infrared (NIR) light, which penetrates deeper into biological tissues, has been shown to improve the efficacy of photodynamic therapy (PDT) by utilizing agents that absorb light in the NIR range, such as ICG. Furthermore, advancements in nanotechnology have led to the development of composite materials that enhance the stability and therapeutic effects of ICG, addressing challenges such as rapid clearance from the body and low specificity in tumor targeting. These innovations highlight the potential of integrating various imaging modalities and therapeutic approaches for improving tumor detection and treatment outcomes [17,18,19].

The aim of this paper is to provide a comprehensive overview of OCT and its evolution, technical underpinnings, and potential future advancements, encompassing all aspects of OCT, from hardware to artificial intelligence, and presenting practical applications in the field. To achieve this, the paper is structured into seven sections. Section 1 serves as an introduction to the topic. Section 2 reviews the history of OCT development, highlighting key milestones. Section 3 delves into the technical foundations of OCT systems, while Section 4 explores the integration of artificial intelligence into OCT technology. Section 5 examines a range of applications for OCT, illustrating its versatility. Section 6 addresses current challenges and limitations and proposes avenues for improvement. Finally, Section 7 concludes the paper, summarizing key insights and outlining prospects for future research and innovation.

## 2. History and Development

This section highlights a selection of notable and highly cited works in the field of OCT, spanning from its inception to the present day. These references provide a foundation for becoming more familiar with the subject literature. OCT was first introduced in the early 1990s by Huang et al. [20]. Images of the eye for examination can be captured non-invasively through cross-section microscopy imaging with OCT using low-coherence interferometry light. This process is similar to ultrasound imaging, in which sound waves are used; in this case, light is used, which makes OCT suitable for imaging the retina and other structures in the eye, and also for determining the structure of plaque deposits in the arteries, which is important for evaluating sickness of the cardiovascular system.

The authors describe the progress and expected use of OCT, as well as its key scientific attributes, spatial resolution, and sensitivity. The system employs fiber optic Michelson interferometry—a simpler system model equipped with a zero-pitch LED light source to blast light into tissues and capture the scattering echoes. The capability of the method to distinguish between different types of tissue and imaging through a variety of scattering media makes the method suitable for a variety of medical research and clinical diagnostic challenges.

A paper published by Swanson et al. in 1993 [21] holds significant importance, as it marked a breakthrough in the practical application of OCT technology. This paper discusses the engineering and clinical evaluation of an in vivo imaging prototype that uses OCT in measuring the human retina below the pupil. The system operates with a ~175 µW at approximately ~843 nm, without exceeding the safety limit (ANSI Z136). It has a ±7 um axial resolution, and a 160 mm/s scanning speed, which is considerably higher than other metrics which achieve a speed of 40 mm/s. It has been demonstrated that the OCT system can be combined with an ophthalmic slit-lamp for precise two-dimensional imaging of the retina.

The authors indicate the clinical usefulness of OCT by showing retinal tomographs and retinal images, including those of the macula and optic disc. The above images show detailed retinal structures, such as the retinal nerve fiber layer (RNFL), choroid, and optic disk profiles, among others. The paper also addresses the motion artifact correction procedures within the in vivo images, which are very important.

In a subsequent study published in 1995 by Hee et al. [22], the researchers demonstrated analogous applications in imaging the human retina with enhanced detail. This research was conducted in a laboratory setting, utilizing a convenience sample of normal human subjects to evaluate the effectiveness of OCT image transfer in retinal imaging.

The OCT used a diode light source which was superluminescent, as well as a fiber optic Michelson interferometer, to acquire depth data from the retina. The sensitivity of the system was such that low-intensity signals could be received, allowing for high-resolution tomographic imaging to be performed.

The OCT results showed a detailed description of the morphology of important anatomical tissues, such as the fovea and the optic disc area, as well as the multilayer retinal structure at a 10 µm level of depth precision. It could even detect benign variants of the retina and retinal nerve fiber layer (RNFL) thicknesses as well. Eye imaging with OCT seems to be useful regarding high-resolution imaging of the fundus, which may assist in the diagnosis and management of retinal diseases such as glaucoma, amoebic macular degeneration, and macular edema.

A paper published by Izatt et al. in 1996 [23] presents significant advancements in optical imaging techniques, particularly OCT and Optical Coherence Microscopy (OCM). OCT is highlighted for its ability to perform high-resolution cross-sectional imaging in biological systems, achieving a resolution that is ten times greater than that of traditional intravascular ultrasound, which is crucial for identifying atherosclerotic lesions that are prone to rupture. The development of a single-mode fiber-optic catheter for OCT allows for the imaging of internal organ systems that were previously inaccessible; its application is demonstrated in imaging a human saphenous vein with enhanced resolution and tissue differentiation. Furthermore, OCM has shown improved optical sectioning depth in highly scattering tissues, enabling the identification of individual crypt cells in colon samples down to a depth of 600 µm, which is a significant improvement compared to conventional confocal microscopy.

In 2000, Rogowska et al. [24] presented the Rotating Kernel Transformation (RKT) technique as an effective method for enhancing OCT images, particularly in the context of coronary plaque detection. The RKT algorithm significantly improves image quality by reducing speckle noise and enhancing the contrast-to-noise ratio (CNR), with the results showing a notable increase in CNR from 1.16 to 6.3 for media-intima regions and from 1.51 to 10.23 for media-plaque regions after processing. Qualitative assessments indicate that processed images exhibit smoother textures and better-defined boundaries of arterial structures, which are crucial for accurate interpretation. Furthermore, the study highlights the importance of kernel size and thickness in optimizing the RKT technique, demonstrating that larger kernels and appropriate thicknesses lead to enhanced boundary detection while preserving the overall shape of profiles.

In another study conducted in 2004, presented by Jain et al. [25] a novel two-axis electrothermal micromirror designed for endoscopic OCT was presented, achieving significant advancements in imaging technology. The micromirror, measuring 1 mm^2^, is fabricated using a deep reactive ion etch post-CMOS process, allowing for large rotation angles of up to 40° and high scanning speeds, which are essential for efficient tissue imaging. The device was demonstrated to have a simple fabrication process compatible with CMOS technology, enabling integration with control circuits on the same chip, thus enhancing its applicability in medical imaging systems. Additionally, the micromirror’s performance was characterized through various experiments, including static and frequency response tests, revealing a resonant frequency of 445 Hz and long-term reliability, with a minimal angular drift of 0.8° over 2 million cycles, indicating its robustness for clinical applications.

In 2007, Salinas et al. [26], presented a comprehensive evaluation of a non-linear complex diffusion approach for enhancing and denoising OCT images, demonstrating significant improvements compared to traditional methods such as the Perona–Malik (PM) filter. The complex diffusion method, which integrates the diffusion equation with the Schrödinger equation, effectively reduces speckle noise while preserving critical image features, achieving an average signal-to-noise ratio (SNR) improvement of approximately 2.5 times and a contrast-to-noise ratio (CNR) enhancement of 49% without degrading the mean structure similarity index (MSSIM). The results indicate that the complex diffusion filter not only outperforms the PM filter in terms of quantitative metrics, but also enhances visual quality, making it particularly suitable for medical imaging applications. Furthermore, the imaginary part of the filtered image can be utilized for guiding segmentation tasks, showcasing the dual functionality of the complex diffusion approach in both denoising and feature extraction.

In 2009, Garvin et al. [27] introduced a significant contribution by developing an innovative extended graph-based approach for segmenting multiple surfaces in 3D spectral-domain OCT images, which incorporates varying feasibility constraints and true regional information, enhancing the utility of layered graph-based segmentation methods. This automated segmentation technique specifically targets the retinal layers, providing significant advancements for the ophthalmology community by enabling the analysis of large volumetric datasets. Additionally, the methodology demonstrated a high level of accuracy, achieving an overall mean unsigned border positioning error comparable to that of two ophthalmologists, thus validating its effectiveness in clinical applications.

A paper published in 2012 by Wilkins et al. [28] presents a fully automated method for segmenting and quantifying cystoid macular edema (CME) from OCT image stacks, demonstrating an average sensitivity of 91% and specificity of 96% in identifying cystoid regions in patients with vitreoretinal disorders. The algorithm effectively computed the total volume occupied by cystoid fluid, achieving a mean error of only 1.9% and a median error of 0.8% when compared to manual inspection, indicating its accuracy in assessing cystoid fractional volume. Additionally, the method incorporates a computationally efficient bilateral filter for speckle denoising, which preserves the boundaries of the CME while significantly reducing the processing time to approximately 2.6 min per image stack.

In 2015, a reported work by Shi et al. [29] presented a novel unsupervised method for automated segmentation of retinal layers in spectral-domain OCT (SD-OCT) images, specifically targeting eyes with serous pigment epithelial detachments (PEDs). The proposed framework includes fast denoising and B-scan alignment, followed by a multi-resolution graph search for surface detection, which significantly enhances segmentation accuracy compared to existing methods, achieving mean unsigned border positioning errors that are statistically indistinguishable from inter-observer variability. The algorithm also effectively detects and segments the PED volume, yielding a true positive volume fraction of 87.1% and a low false positive volume fraction of 0.37%, demonstrating its robustness in distinguishing normal retinas from those with PEDs. Furthermore, the method is computationally efficient, with an average running time of 220 s for processing OCT data, making it suitable for clinical applications.

Fang et al. [30], in 2017, presented a novel method called Segmentation-Based Sparse Reconstruction (SSR), aimed at improving the quality of retinal OCT images, which are often affected by noise and low spatial resolution. By utilizing a segmentation algorithm, the SSR method divides OCT images into distinct layers, each containing specific anatomical and pathological features, allowing for the construction of tailored dictionaries that enhance image reconstruction performance. This approach not only improves the denoising and interpolation of OCT images, but also significantly reduces computational costs compared to traditional methods that search the entire image for similar patches. The experimental results demonstrate that the SSR method outperforms several existing techniques, making it a promising tool for enhancing the analysis of OCT images in medical applications.

In 2019, Zaiwang Gu et al. [31] presented CE-Net, a novel deep learning framework designed for medical image segmentation, which is crucial for analyzing medical images like those of the retina or lungs. This method improves upon traditional techniques by using a combination of advanced components, including a feature encoder, a context extractor, and a feature decoder, to better capture and preserve important details in images. By integrating specialized blocks that enhance feature extraction and pooling, CE-Net has demonstrated superior performance in various segmentation tasks, such as optic disc and vessel detection, compared to existing methods like U-Net. The results indicated that CE-Net not only enhances accuracy, but also provides a more efficient approach to medical image analysis, making it a significant advancement in the field.

In 2019, another work by Fang et al. [32] presented the Lesion Aware Convolutional Neural Network (LACNN), a novel approach designed to enhance the classification of retinal images obtained through OCT, which is essential for diagnosing various eye diseases. This method leverages the attention mechanism to focus on specific areas of the images that are indicative of lesions, thereby improving the accuracy of the classification process 16. The LACNN operates by first detecting these lesions and then using this information to guide the classification network, allowing it to prioritize relevant features while minimizing the influence of less important data. Experimental results have shown that LACNN significantly outperforms traditional methods, demonstrating its effectiveness in clinical settings for eye disease diagnosis. Table 1 illustrates the timeline of Optical Coherence Tomography (OCT) development.

## 3. OCT System Technology

OCT device is a sophisticated imaging system that consists of three main components: optical, electronic, and software. Each of these components plays an important role in the overall functionality of the device, enabling it to capture high-resolution cross-sectional images of biological tissues. The integration of these elements allows OCT to provide detailed insights into the microstructure of tissues, making it an invaluable tool in various medical applications.

### 3.1. Optical Part

The optical section of an OCT device is composed of four elements: a broadband light source, a splitter, a reflector, and a sensor.

Typically, an SLD (Superluminescent Diode) is used as the light source in an OCT device. SLDs generate light with low coherence, which implies a high frequency bandwidth and a short coherence length. The high bandwidth of the light source helps to eliminate ambiguity in distance measurement [33,34,35].

The splitter typically has two functions: firstly, it divides the emitted light into two parts, and secondly, it combines the two reflected beams of light. Splitters are predominantly composed of glass or plastic which is transparent [36,37,38].

The reflector or mirror has the function of reflecting light. The mirrors must be composed of materials that can reflect light in the required range for OCT (usually near-infrared). Fabrics like aluminum, silver, and certain alloys of the above-mentioned materials are common stables. These materials have anti-scratch and protective coatings to withstand the external environment and improve performance [39,40].

Photodetector sensors or interferometric cameras are used as sensors in OCT devices. These sensors capture the light that has interacted with the sample and the reference beam, facilitating the interference pattern analysis that is essential for constructing high-resolution images of the tissue [41,42].

#### 3.1.1. Operational Basis of OCT Optical Part

The operational principles of OCT are very simple. Light comes from a light source, and this light travels through a beam splitter, which splits it into two optical paths. One of them travels through a reference mirror, and the other one goes to the tissue that needs to be examined. The irradiance that comes back from the mirror, as well as the tissue, is reflected into the beam splitter, where the two irradiance beams are simultaneous recombined.

The interference that evolves from this recombination is then focused onto a detector or a sensor, allowing depth-resolved images to be acquired, making it possible to obtain interferometric images, also known as tomograms. The process is demonstrated in Figure 1.

#### 3.1.2. Physical and Quantitative Examination of Operations

The stages that have been described so far represent a simplified qualitative operation that forms the foundation of OCT. However, to implement OCT, it is necessary to understand the quantitative physical relationships involved, which are further discussed in the subsequent sections.

The optical path difference between the sample and reference paths, necessary for creating the interference, is calculated as shown in Equation (1) [43], where *n* is the refractive index of the medium through which light travels, and *d* is the optical path length traveled by the light within the medium.(1)OPT=2(n.d)

The intensity of the interference light detected can be described by the interference equation shown in Equation (2) [44], where *I*_1_ and *I*_2_ are the intensities of the light reflected from the sample and reference paths, respectively, and Δ∅ is the phase difference between the two light waves.(2)$$I=I_{1}+I_{2}+2\sqrt{I_{1}I_{2}}\cos\left(\Delta \emptyset \right)$$

The axial resolution, which determines the imaging accuracy in the depth direction, is calculated according to Equation (3) [43], where *n* is the refractive index of the medium.(3)$$Axial\;Resolution=\frac{\lambda^{2}}{2n\Delta \lambda }$$

The lateral resolution is dependent on the diameter of the focused light spot, and can be calculated using Equation (4) [44], where *NA* is the numerical aperture of the optical system.(4)$$Lateral\;Resolution=\frac{0.61\cdot \lambda }{NA}$$

In Figure 1, a simple schematic of the OCT optical part is presented.

#### 3.1.3. Various Types of OCT

The described structure is merely a simplified model of OCT; actual models possess many complexities. For example, OCT can be categorized into various types based on structural differences. Below are some of these types.

Time Domain OCT (TD-OCT) works by measuring the time delay of light reflected from different depths within the sample using a reference mirror and a low-coherence light source. TD-OCT can identify different layers in the retina, helping to diagnose diseases such as glaucoma and macular degeneration [45].

Spectral Domain OCT (SD-OCT) uses a broadband light source and a spectrometer. Unlike TD-OCT, SD-OCT measures the interference spectrum of reflected light, allowing for faster image acquisition and improved resolution. This technology is particularly useful in ophthalmology for examining the retina and diagnosing conditions like macular degeneration and diabetic retinopathy. SD-OCT’s ability to provide real-time, high-resolution images makes it a valuable tool in clinical practice [46,47].

Swept Source OCT (SS-OCT) uses a tunable laser source to capture high-speed and high-resolution cross-sectional images of biological tissues. By sweeping the laser across a range of wavelengths, SS-OCT collects depth information more quickly than traditional methods. This technology is particularly beneficial in ophthalmology, enabling detailed visualization of the retina and choroid, and is effective in diagnosing and managing eye diseases like glaucoma and age-related macular degeneration. SS-OCT’s extended imaging depth and speed make it suitable for a wide range of clinical applications [48,49].

Fourier Domain Mode-Locked OCT (FDML-OCT) is an advanced imaging technique that utilizes an FDML laser to achieve high-speed and high-resolution imaging. FDML lasers enable rapid wavelength sweeping, allowing for fast acquisition of OCT images without the need for resampling in the frequency domain. This technology significantly enhances imaging speed and depth range, making it particularly useful in medical imaging applications such as ophthalmology. FDML-OCT provides detailed cross-sectional images of tissues, aiding in the diagnosis and monitoring of various conditions. The combination of high speed and resolution makes FDML-OCT a valuable tool for real-time imaging applications [50,51,52]. A qualitative comparison between different OCT technologies is given in Table 2, and Figure 2 also illustrates the various types of OCT.

## 4. Electronics

An essential aspect of the OCT system is its electronic component, which facilitates the processing of optical signals and their conversion into digital images. The above process starts with the use of detectors, usually either photodiodes or other photodetectors, which, through the photoelectric effect, harness light energy to create a current, forming the basis of operation of the device [41,42].

The electrical signals produced by the photodetectors are classified as weak signals, and therefore require amplification. Amplifiers are crucial in addressing this limitation by strengthening the signals to a level that allows central processing systems to accurately interpret the incoming data [53].

Filters are employed to suppress interference and enhance the quality of the electrical signals by eliminating non-optimal frequencies and background noise, thereby improving the accuracy of the final image [54].

Once a signal has been amplified, it is essential to digitize it for computer-based operations. This function is performed by Analog-to-Digital Converters (ADCs), which enable advanced digital data processing and analysis [54,55].

In an OCT system, the control mechanism is responsible for configuring the device parameters, operating moving parts such as scanning mirrors, and adjusting the imaging settings [56,57]. Additionally, software applications may be integrated for data interpretation and image rendering within the defined system. The processing unit facilitates activities such as signal processing, data management, and the operation of the interactive interface [58,59].

In other words, the electronic components of an OCT system work synergistically to provide effective imaging of biological tissues with minimal noise, which is invaluable in medical diagnostics. The integration of these components ensures that OCT can generate high-quality images in a time-sensitive manner, which is critical for both clinical and practical applications.

### 4.1. Software

The software of the OCT system is responsible for comprehensive signal processing, which includes the extraction of A-scan images, the assembly of these images into B-scan images, and further synthesis into three-dimensional C-scan images. Ultimately, artificial intelligence algorithms are employed to process the derived images, ensuring that both the images and analytical results are efficiently transmitted to the user.

In other words, the software plays a crucial role in managing the entire imaging process, from data acquisition to final analysis. It begins by extracting A-scan images, which represent the depth and intensity of signals at specific points within the sample. These A-scans are then meticulously assembled into B-scan images, providing a cross-sectional view of the sample. This step is essential for visualizing the internal structures with high precision. Furthermore, the software synthesizes multiple B-scans into three-dimensional C-scan images, offering a comprehensive volumetric representation of the sample. This three-dimensional perspective is invaluable for detailed analysis and diagnostic purposes. The integration of artificial intelligence algorithms enhances the processing capabilities, enabling the software to perform advanced image analysis, such as noise reduction, feature extraction, and pattern recognition. These AI-driven processes ensure that the final images and analytical results are of high quality and accuracy, facilitating efficient transmission to the user for further evaluation and decision making. The software’s ability to handle large datasets and perform complex computations in real time makes it an indispensable tool in both clinical and research settings, ultimately contributing to improved diagnostic outcomes and patient care.

#### Signal Processing

Extracted digital data are fed into the processor, where signal processing operations are performed on these data. Initially, the data enter a preprocessing stage, in which noise levels are corrected using filters. The first part of signal processing is background removal. Background removal in OCT signal processing is a critical step to enhance image quality and accuracy. It involves eliminating unwanted components, such as DC offset, system noise, and spurious reflections, that can obscure meaningful information. By subtracting the average signal (obtained from reference or baseline measurements) or applying advanced filtering techniques, the process improves the SNR and enhances contrast.

Subsequently, the main processing operation, which is the Fourier Transform, is performed on the data. The Fourier Transform is essential for converting overlapping signals into spatial data. The use of the Fast Fourier Transform (FFT) is necessary as it enables the observation and analysis of the amplitude and phase of signals in frequency space [60,61].

The A-scan results from the Fourier Transform are displayed on a graph, where the horizontal axis represents depth and the vertical axis represents signal intensity. This graph aids in the precise localization of structures within the sample. To produce a B-scan image, the optical scanner moves laterally across the sample, generating an image line (A-scan) at each point. These consecutive lines are joined together to form a two-dimensional image. To conclude, building a three-dimensional representation that is regarded as a C-scan requires joining together various B-scans. Repetitive B-scans define the shape of the volumetric images of the analyzed specimen in order to study its internal structure in greater detail.

## 5. Artificial Intelligence Applied to OCT

Artificial intelligence (AI) is not inherently a component of OCT systems. However, integrating AI with OCT can lead to substantial improvements in signal quality and diagnostic outcomes. AI’s contributions to OCT systems encompass multiple facets: it can enhance image quality through advanced algorithms that reduce noise and improve resolution, thereby enabling clearer visualization of tissue. Additionally, AI can play an important role in disease detection by automating the analysis of OCT images, facilitating the identification of pathological changes that may be indicative of conditions such as diabetic retinopathy or age-related macular degeneration. Furthermore, AI can assist in processing images for more accurate assessments and predictions regarding disease progression, ultimately guiding clinical decision making and improving patient management [62,63,64].

### 5.1. Deep Learning

Deep learning (DL) algorithms are extensively used in OCT for various applications, including disease differentiation, feature segmentation, and image quality assessment. These algorithms leverage neural networks to analyze and interpret OCT images, providing valuable insights for clinical diagnosis and management.

Disease Differentiation: DL models are highly effective in distinguishing between different diseases. For instance, they can differentiate between normal and diseased tissue structures, such as those affected by age-related macular degeneration (AMD) or diabetic retinopathy (DR). This capability is very important for early diagnosis and timely intervention in tissues [65,66].

Feature Segmentation and Quantification: DL algorithms excel in segmenting and quantifying specific features within OCT images. Techniques like U-Net and other Convolutional Neural Networks (CNNs) are used to accurately delineate layers and identify changes. This segmentation is essential for the precise measurement and monitoring of disease progression [67,68].

Image Quality Assessment: DL models can also evaluate the quality of OCT images, ensuring that the data used for diagnosis are reliable. This is particularly important in clinical settings where image quality can significantly impact diagnostic accuracy [69,70].Cross-Sectional Dataset Analysis: DL algorithms are adept at analyzing cross-sectional datasets, which are common in OCT imaging. This allows for the identification of diseases across different patient populations and imaging modalities, enhancing the generalizability of diagnostic tools [71,72].Non-Invasive Cancer Diagnosis: DL has been applied to predict histological images directly from OCT images, providing a non-invasive method for cancer diagnosis. This application showcases the potential of DL in advancing medical imaging beyond traditional OCT uses [73,74].

In other words, deep learning algorithms in OCT are pivotal for enhancing diagnostic accuracy, automating feature extraction, and improving the overall quality of imaging. These advancements contribute significantly to the early detection and management of diseases [67,75,76].

#### 5.1.1. Transfer Learning

Transfer learning in OCT leverages the knowledge gained from training deep learning models on large, diverse datasets and applies it to the specific task of analyzing OCT images. This process involves using a pre-trained model which has been trained on a general image dataset like ImageNet, and then fine-tuning it with a smaller, OCT-specific dataset. The pre-trained model provides a robust feature extraction capability, which is then adapted to recognize the unique features of OCT images, such as the layers of the retina and pathological changes associated with diseases like AMD and DR.

The fine-tuning process typically involves updating the weights of the pre-trained model using backpropagation with the OCT dataset. This allows the model to learn the nuances of OCT images while retaining the general image recognition capabilities it acquired during its initial training. Transfer learning significantly reduces the computational resources and time required to train a model from scratch, making it a practical approach for developing AI-based diagnostic tools in ophthalmology [77,78].

Moreover, transfer learning helps in overcoming the challenges posed by the limited availability of labeled OCT data. Since OCT images can be complex and require expert knowledge to label accurately, transfer learning enables the utilization of existing models that have been trained on extensive datasets, thus improving the accuracy and reliability of OCT image analysis. This approach has been successfully applied in various studies to detect and classify retinal diseases, demonstrating its potential to enhance clinical decision making and patient care [79,80].

#### 5.1.2. CNNs

CNNs are a class of deep learning algorithms specifically designed for processing data that have a grid-like topology, such as images. In the context of OCT image analysis, CNNs are particularly effective due to their ability to automatically and adaptively learn spatial hierarchies of features from raw OCT images.

CNNs consist of multiple layers, including convolutional layers, pooling layers, and fully connected layers. The convolutional layers perform the core operation of CNNs, which is the convolution of the input image with a set of learnable filters (also known as kernels). These filters slide over the image to capture local patterns and features, such as edges, textures, and shapes. The pooling layers, typically max-pooling or average-pooling, down-sample the feature maps produced by the convolutional layers, reducing the dimensionality of the data and making the network more robust to small variations in the input.

The fully connected layers at the end of the network aggregate the high-level features extracted by the convolutional and pooling layers to perform the final classification or regression task. For instance, in OCT image analysis, CNNs can be trained to segment the retinal layers, detect the presence of fluid or lesions, and classify diseases such as AMD or DR.

One of the key advantages of CNNs is their ability to learn hierarchical representations of data. Lower layers in the network learn simple features like edges and textures, while deeper layers learn more complex, abstract features that are specific to the task at hand. This hierarchical structure allows CNNs to achieve high accuracy in image recognition and classification tasks, making them a powerful tool for OCT image analysis.

In summary, CNNs are well suited for OCT image processing due to their ability to automatically learn relevant features from raw image data, their robustness to variations in the input, and their hierarchical structure, which enables the learning of complex patterns and relationships within OCT images [81,82].

#### 5.1.3. Support Vector Machine (SVM)

SVM is a powerful machine learning algorithm used for classification and regression tasks. In the context of OCT image analysis, SVM is employed to differentiate between normal and diseased retinal structures, particularly in the detection of glaucoma and AMD.

SVM works by finding the hyperplane that best separates different classes in a high-dimensional space. This hyperplane is determined by the support vectors, which are the data points closest to the decision boundary. SVM can handle both linear and non-linear data by using different kernel functions, such as linear, polynomial, radial basis function (RBF), and sigmoid kernels. The choice of kernel function depends on the nature of the data and the problem at hand.

In OCT image analysis, SVM is used to classify OCT images by extracting relevant features from the images and then applying the SVM algorithm to these features. For instance, SVM has been successfully applied to Spectralis OCT to differentiate glaucomatous from normal eyes, demonstrating good diagnostic capability. Additionally, SVM has been used in conjunction with 3D OCT volumes to assist in the detection of AMD, further highlighting its versatility and effectiveness in ophthalmic applications [83,84].

The use of SVM in OCT image analysis offers several advantages, including high accuracy, robustness to overfitting, and the ability to handle high-dimensional data. These characteristics make SVM a valuable tool in the development of automated diagnostic systems for retinal diseases, enhancing the precision and efficiency of clinical decision making [85,86]. Table 3 provides a summary of the applications of AI in OCT.

## 6. Clinical Applications of OCT Imaging

After gaining an understanding of OCT, it is essential to explore its diverse applications. This section focuses on notable studies involving OCT, highlighting their specific applications, the results obtained, and the outcomes achieved.

In a study conducted by Gardecki et al., micro-OCT was presented [87]. µOCT offers high-resolution imaging of prostate tissue, and is capable of resolving architectural and cellular features associated with benign and neoplastic prostate conditions. The µOCT system uses spectral domain OCT with a broad-bandwidth light source, achieving an axial resolution of less than 1 µm. Despite its penetration depth of 300–500 µm, which is greater than other in vivo microscopy modalities, it remains insufficient for imaging the entire prostate with a realistic number of needle insertions. The study suggests that imaging at longer wavelengths could significantly increase penetration depth, potentially doubling or tripling current values. Implementing µOCT in a small diameter probe for in vivo use could reduce biopsy sampling errors and enhance prostate cancer diagnosis. The study found that 14% of the samples contained prostate cancer, while 86% were benign, indicating µOCT’s potential utility in prostate diagnostics [87].

In another study conducted by Zhou et al., the research focused on detecting early-stage degeneration of human articular cartilage using polarization-sensitive OCT (PS-OCT) [88]. The study demonstrated the efficacy of PS-OCT in differentiating bone tissue types, particularly in the context of prostate cancer-associated bone metastases (PCBM). The degree of ordered organization (DOO) feature, derived from PS-OCT, effectively distinguishes between trabecular and irregular bone regions, offering a non-invasive alternative to traditional imaging methods. The integration of PS-OCT with MATLAB-based image analysis tools allows for detailed examination of bone microstructures, such as lacunae morphology, at a rapid pace, enhancing the efficiency of research cycles. Despite its potential, the study acknowledges the sensitivity of PS-OCT to environmental conditions and the need for further development for in vivo applications. The combination of PS-OCT with conventional techniques like CT and SEM provides a comprehensive understanding of bone tissue dynamics, paving the way for advancements in bone-related pathologies. Future work aims to develop a fiber-based endoscope for in vivo imaging, potentially expanding the clinical utility of PS-OCT [88].

In a paper presented by Waheed et al. [89], OCT-Angiography (OCTA) emerged as a pivotal tool in the assessment of diabetic retinopathy (DR), offering detailed insights into retinal microvascular changes. The technology’s ability to visualize and quantify the foveal avascular zone (FAZ) and vessel density means that it can provide critical data for evaluating DR severity and progression. Despite its potential, challenges such as image artifacts and variability in FAZ metrics remain, necessitating further refinement in imaging techniques and standardization across studies. Recent advancements in OCTA, including higher-speed platforms and improved software, are enhancing the precision of peripheral retina assessments, which are crucial for detecting neovascularization and non-perfusion. These developments underscore OCTA’s role in advancing our understanding of DR and supporting the development of new therapeutic strategies. Future research should focus on correlating OCTA metrics with visual functions to validate their clinical utility.

In another work presented by Azzollini et al. [90], Dynamic-OCT (D-OCT) was presented as a cutting-edge imaging technique that provides label-free, live optical imaging of dynamic cellular and subcellular features by analyzing temporal fluctuations of optical signals associated with intracellular organelle movements. This method offers insights into cellular physiology, and is particularly promising for the three-dimensional evaluation of live tissue samples, such as freshly excised biopsies and 3D cell cultures. D-OCT leverages the temporal behavior of optical signals to gain deeper insights into cellular mechanisms, allowing for the visualization of specific cells and their nuclei, and identifying the mitotic states of cells. It circumvents the need for fluorescent dyes, thus avoiding phototoxicity and biases introduced by fluorescent markers. Applications of D-OCT include monitoring cell states, detecting apoptosis, and assessing responses to anti-cancer drugs in vitro. It is also used for ex vivo tissue analysis, such as detecting different tissue components and changes in OCT signal variance due to lipid droplet movements.

Huang et al. presented a needle-based OCT technique with real-time visualization capabilities [91]. The study investigated the use of Needle-Probe OCT for real-time visualization of Veress needle placement in a porcine model, aiming to enhance the safety of pneumoperitoneum establishment in laparoscopic surgery. The primary outcome was a 97.5% success rate in peritoneal punctures, with no intra-abdominal organ injuries reported. The OCT system transformed the traditionally blind closed technique into a visualized procedure, improving the safety of peritoneal access. Statistical analysis showed a significant difference in the standard deviation (STD) of the OCT images, indicating a high discrimination capability between the peritoneum and extra-peritoneal tissue, with an area under the ROC curve (AUC) of 0.97. The study suggests that the OCT system could be a valuable tool for minimally invasive procedures in modern surgery. Additionally, 74.7% of surveyed surgeons expressed a willingness to use the Veress needle again if an assistance device could visualize the puncturing process.

Another work by Kuo et al. [92] highlighted the effectiveness of the Quadratic Support Vector Machine (QSVM) classifier in identifying the epidural space (ES) with high sensitivity (97.5%), specificity (95%), and accuracy (96.2%). The OCT needle probe, integrated with machine learning, provides real-time, high-resolution imaging for accurate needle placement in medical procedures, particularly in neuraxial blocks. The handheld OCT needle probe, although currently limited to larger needles, offers potential for compact and cost-effective clinical applications. The system’s ability to provide detailed anatomical imaging from the needle tip complements ultrasound-guided methods, although it does not guide the needle’s trajectory. The OCT technology was also explored for other medical applications, such as fascial blocks, emphasizing the need for further validation in clinical practice. The integration of OCT with AI presents the opportunity to improve the quality of medical care amidst increasing demand and limited human resources. Table 4 presents a comparison of the works conducted.

## 7. Challenges and Limitations

OCT faces a variety of challenges across the domains of optics, electronics, AI, and software development.

From an optical perspective, it is necessary to design a light source that possesses appropriate coherence length and wavelength. Additionally, stabilizing the interferometer and controlling beam delivery and scanning techniques are essential for achieving high-resolution imaging.

From an electronic standpoint, acquiring high-quality images necessitates rapid data acquisition and precise timing coordination. Therefore, the electronic components must be optimized to handle the high-speed data transfer required for effective imaging.

In terms of software, there is a significant need to develop efficient image reconstruction algorithms, as well as user-friendly graphical user interfaces (GUIs) and robust database management systems tailored to clinical environments. These software components must ensure that the imaging data can be easily accessed and interpreted by healthcare professionals.

Regarding AI and machine learning, several major challenges persist, including the collection of annotated datasets, the development of sophisticated yet interpretable models, and the integration of AI technologies with OCT systems in a manner that meets regulatory approval. Addressing these challenges comprehensively is vital for advancing OCT technology and enhancing its clinical applications.

### 7.1. Optical Challenges

OCT presents several interesting challenges that require detailed attention and improvement. One of the primary challenges is the design of the light source. To achieve high axial resolution, it is essential to utilize Superluminescent Diodes (SLDs) or swept source lasers, as these light sources provide the necessary long coherence length. Such light sources enable reliable resolution of small features within biological tissues. Furthermore, selecting the appropriate wavelength is critical, as different tissues demonstrate varying responses to light. By carefully choosing the wavelength range, it is possible to optimize contrast against the tissue background and enhance penetration depth into the target tissue.

Another significant consideration in OCT systems is the stability of the interferometer. The system must possess mechanical stability to maintain alignment and coherence throughout the imaging process. Mechanical vibrations or thermal fluctuations can introduce errors, leading to degraded image quality. In addition to mechanical stability, polarization control stability is vital. Effective polarization control allows for high interference contrast; conversely, a loss of polarization control can result in substantial signal degradation, negatively affecting both image quality and the reliability of the OCT system.

The en face modality of OCT is heavily dependent on the beam delivery system and the scanning mechanisms employed. To obtain high-resolution images, these scanning mechanisms must operate at high speeds while ensuring accuracy. Techniques such as galvanometer scanners or micro-electromechanical systems (MEMS) are commonly utilized. However, challenges also arise in focusing and beam shaping, particularly when imaging larger specimens. Achieving uniform beam intensity and a consistent focal point across the entire field of view is a complex task that necessitates advanced optical engineering solutions. Such engineering innovations are essential to enable high-quality imaging of entire specimens, thereby facilitating the extraction of meaningful structural information.

### 7.2. Electronics Challenges

The resolution of electronic and optical challenges in functional OCT devices is equally important, particularly with respect to data acquisition speed. The implementation of Analog-to-Digital Converters (ADCs) is critical, as these devices must achieve high speed and resolution to facilitate real-time data capture. For effective acquisition of the minute interferometric signals that are necessary for imaging, these converters must operate within minimal bandwidth constraints and sample at a high frequency. Such precision is essential to preserve the finer details of the image, enabling effective analysis and diagnosis.

In addition to ADCs, the processing of weak OCT signals necessitates effective signal amplification and conditioning. It is crucial that these signals are amplified to a usable level without introducing noise that could obscure subtle image features. Balanced detectors and Low-Noise Amplifiers (LNAs) play a vital role in enhancing signal strength while maintaining signal integrity. This balance is important for ensuring that the resultant images are not only more pronounced, but also richer and clearer, which is essential for any medical or scientific application.

Synchronization is another significant challenge faced by OCT systems. The timing coordination among the light source, scanning mechanisms, and data acquisition instruments must be meticulously controlled. This is particularly critical in Swept Source OCT systems, in which misalignments can yield erroneous results. Effective management of the sequential operations ensures that all the components of the system integrate seamlessly, thereby enhancing the accuracy of the captured images.

Moreover, clock jitter is a well-known issue in the electronic components of OCT systems. Jitter refers to small, rapid deviations from the average or nominal timing of a clock signal, which results in phase shifts in the referenced interferometric signals. Such timing errors can degrade image quality, making it imperative to minimize jitter to the lowest levels possible. By addressing these synchronization issues, OCT systems can achieve accurate imaging across all targeted areas, which is vital for their effective use in clinical and research settings.

### 7.3. Software Challenges

In the domain of OCT, one of the significant challenges is the transformation of raw data into clinically relevant images that are user friendly and can be seamlessly integrated into clinical workflows. The development of advanced algorithms for image reconstruction is a primary concern in this regard. The objective of these algorithms is to efficiently process signals to generate high-quality images. This involves several sophisticated techniques, including spectral shaping, which enhances image quality by modifying the spectral profile, and dispersion compensation, which corrects for the varying rates at which light travels through different tissues. Additionally, noise reduction techniques are essential for eliminating artifacts, thereby ensuring that the resulting images are diagnostic quality and free from obstructions, which is critical for effective diagnostic imaging.

In clinical applications, where time is of the essence, images must be processed in real time. This necessitates that wirelessly connected devices integrate seamlessly into existing systems. Consequently, algorithms must be optimized for rapid processing while still maintaining high image quality. A key challenge is to ensure that the time required for capturing an image or performing an environmental scan remains within acceptable limits without compromising detail or accuracy. This optimization is vital for making OCT systems practical and efficient for real world applications.

Furthermore, the implementation of IUCH (International Universal Classification of Health) levels 3 and 4 presents substantial software engineering challenges beyond the technical aspects of image reconstruction. Specifically, the user interface and data management require careful consideration. A user-friendly interface is important in clinical settings, allowing healthcare professionals to quickly review and adjust imaging parameters. Enhancing the usability of the software can significantly improve the application and efficiency of OCT systems, facilitating easier access to imaging data for clinicians.

Data storage and retrieval are also critical due to the vast amounts of data generated by OCT systems. There is a pressing need for efficient data management systems that enable secure storage, retrieval, and integration with Electronic Health Records (EHRs). Such systems should not only ensure the safe storage of imaging data, but also facilitate convenient access to other health information, ultimately improving patient care and providing a more comprehensive approach to patient management. Addressing these software challenges is essential in order to fully realize the potential of OCT technology in clinical practice.

### 7.4. AI and Machine Learning Challenges

The integration of AI and machine learning into Optical Coherence Tomography (OCT) presents numerous challenges. When expanding the capabilities of OCT, several difficulties must be addressed, including the adoption of AI and machine learning, data annotation, infrastructure development, and system integration.

One of the primary bottlenecks is the annotation and labeling of data required for training AI models. The process of creating ground truth data is both intricate and labor-intensive; depending on the model’s complexity, it may necessitate the involvement of skilled professionals. Additionally, the variability of biological tissues introduces complexity, making it difficult for models to be robust and generalizable across diverse patient populations and clinical scenarios. To ensure reliability, models must be trained on a broad and diverse dataset that adequately represents variations in tissue characteristics and patient demographics.

Moreover, the creation and validation of AI models present additional challenges. Developing algorithms that not only generate accurate scans but also interpret them effectively is a complex task. The models must strike a balance between complexity and efficiency; overly complex models may become inefficient, hindering their practical application. This trade-off between model complexity and interpretability is necessary for achieving acceptance in real-world clinical settings. Furthermore, computational efficiency is essential to enable rapid implementation and application of these models, ensuring timely results.

Once AI models are developed, they must undergo rigorous validation against diverse datasets to ensure their clinical applicability and accuracy. The specific integration of AI into OCT systems is fraught with additional challenges, particularly regarding the need for substantial processing power and the establishment of functional interfaces with existing hardware and software. Smooth integration is essential to minimize processing delays and maximize the utility and responsiveness of AI-enhanced OCT systems within clinical contexts.

Additionally, AI-enhanced OCT systems must navigate stringent regulatory approval processes. While these regulations are necessary to ensure the safety and efficacy of the systems, they can be time consuming and costly. Addressing these regulatory barriers is vital for facilitating the transition of AI-driven solutions from research environments to clinical settings, ultimately enhancing diagnostic quality and improving patient treatment outcomes.

## 8. Future and Development

Significant advancements are on the horizon for OCT research and development across several key areas. Enhancing resolution and imaging depth has become a primary focus, with scientists working to develop superior light sources and scanning techniques to improve image quality and enable deeper tissue penetration. Additionally, high-speed imaging is an important area of exploration, aiming to enhance imaging hardware and software while reducing motion artifacts, thereby facilitating real-time imaging capabilities.

There is also a growing interest in utilizing OCT in conjunction with other imaging modalities, such as fluorescence and ultrasound. This multimodal approach allows for a more comprehensive assessment of tissue structures and functions.

In terms of practical applications, OCT remains integral to ophthalmology, where it is widely used for diagnosing and monitoring various retinal diseases. Future research aims to develop more accurate and reliable methods for early disease detection and the evaluation of therapy effectiveness. In cardiology, advancements in intracoronary OCT are focused on obtaining high-resolution images of coronary arteries, which will aid in the management of heart disease.

Research in neurology is investigating the potential of OCT for brain analysis, with an emphasis on developing techniques that are capable of penetrating the cranium to visualize brain structures effectively.

A significant trend in the field of OCT molecular imaging is the integration of AI and machine learning, particularly in predictive analytics. As AI networks are developed for tasks such as segmentation, classification, and quantification, the need for manual analysis will diminish. Predictive models are also being created to estimate disease progression and therapeutic outcomes based on available OCT data, thereby assisting in treatment planning.

Another critical aspect necessary for the modernization of OCT is interdisciplinary collaboration. This involves integrating the expertise of physicists, engineers, clinicians, and computer scientists to enhance the translation of technological advancements into clinical practice. Such collaboration should also encompass legal and ethical considerations to protect emerging OCT technologies from misuse and to ensure respect for privacy and ethical standards in AI applications.

Recent efforts have been directed toward developing simple and cost-effective OCT systems for use in resource-limited settings, with a particular focus on miniaturization and cost reduction. Furthermore, OCT is increasingly being applied in personalized medicine, in which imaging data are utilized to determine the most suitable treatment for individual patients. Algorithms analyze relevant information from databases to provide personalized treatment recommendations, thereby enhancing patient care.

## 9. Conclusions

Stereodimensional imaging (three-dimensional imaging) continues to gain popularity, particularly in healthcare, where OCT has emerged as a significant imaging modality. This essay aims to explore all the features of OCT and to assess its future potential. In recent years, there has been substantial interest in AI applications in healthcare, as AI-centered care systems hold the promise of addressing unmet global healthcare needs and creating opportunities to alleviate shortages within primary workflows.

The synergy between deep learning and biomedicine has made considerable inroads into various scientific domains, leveraging the vast amounts of data and knowledge available while benefiting from automated analysis. However, combining AI and machine learning with OCT presents complex challenges. The feasibility of implementing a deep learning model depends on the development of sophisticated programming frameworks that ensure efficient performance while maintaining interpretability.

Future discussion should focus on three critical concepts that require innovative solutions to create effective AI-driven tools in the context of OCT.

Data Quality and Annotation: High-quality annotated datasets are essential for training deep learning models effectively. Ensuring the availability of comprehensive and accurately labeled data presents a significant challenge.

Model Interpretability: The complexity of deep learning models often leads to difficulties in interpretation, which can hinder clinical acceptance. Developing models that are both sophisticated and easily interpretable is important for their successful integration into clinical practice.

Regulatory Compliance and Validation: The integration of AI into medical imaging systems must comply with regulatory standards. This necessitates thorough validation of AI models to ensure their safety and effectiveness in clinical settings.

Overall, the evolving trends in OCT technology, driven by the integration of AI and cross-disciplinary collaboration, are likely to enhance medical imaging capabilities and improve patient outcomes. By addressing the aforementioned challenges, the potential for AI-driven OCT systems to revolutionize clinical practice becomes increasingly attainable.

## Figures and Tables

**Figure 1 bioengineering-12-00065-f001:**
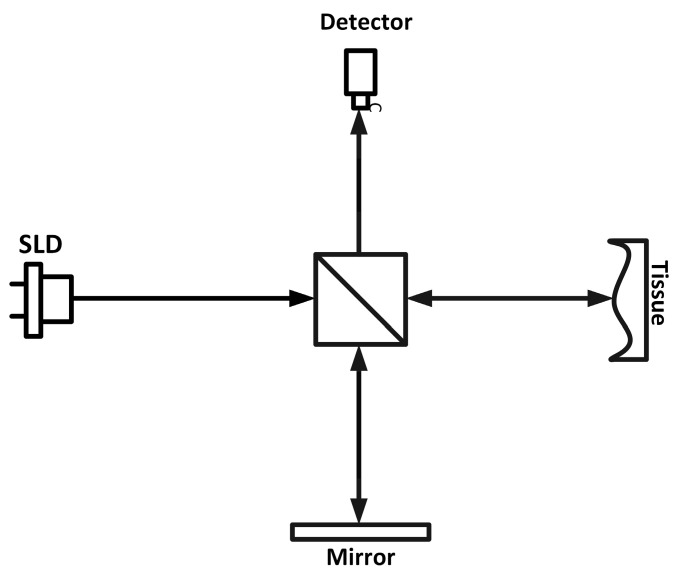
Simple schematic of OCT optical part.

**Figure 2 bioengineering-12-00065-f002:**
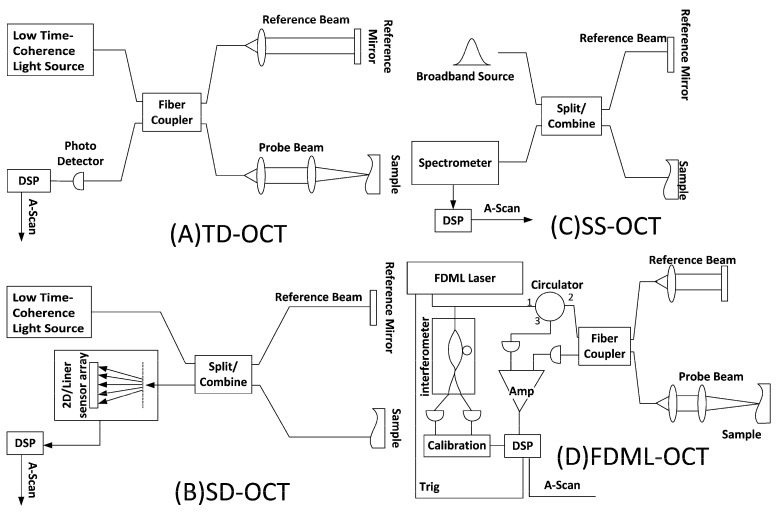
Various types of OCT.

**Table 1 bioengineering-12-00065-t001:** The progress of OCT over time.

Year	OCT Technology	Description
1991	Invention of OCT	OCT was first introduced as a non-invasive imaging technique for biological tissues
1990s	Time Domain OCT (TD-OCT)	TD-OCT was developed, using a moving reference mirror for tissue scanning
2000	Spectral-Domain OCT (SD-OCT)	SD-OCT was introduced, using a spectrometer to measure back-reflected light from tissues
2005	Swept-Source OCT (SS-OCT)	SS-OCT was introduced, using a swept-source laser for tissue scanning
2010	Fourier-Domain Mode-Locking OCT (FDML-OCT)	FDML-OCT was introduced, using FDML lasers for tissue scanning, providing very high speed and image clarity
2015	Combination of SD-OCT and SS-OCT	The combination of SD-OCT and SS-OCT was used to improve image accuracy and depth
2020	Advanced OCT Techniques (OCT-A, Molecular OCT)	Advanced OCT techniques such as OCT angiography and molecular OCT were introduced, enabling imaging of blood flow and specific molecules
2024	New Applications of OCT (Neurosurgery, Occupational Therapy, Dermatology)	OCT was introduced in new fields such as neurosurgery, occupational therapy, and dermatology

**Table 2 bioengineering-12-00065-t002:** Qualitative comparison of OCT technologies.

Attribute	TD-OCT	SD-OCT	SS-OCT	FDML-OCT
Advantages	Longer depth range Simpler technology Reliable for deep tissue imaging	Higher resolution Faster imaging speed Higher SNR Better for real-time imaging	Faster acquisition Better depth penetration No sensitivity roll-off Better for imaging through various media	Extremely high imaging speeds Higher SNR and sensitivity Better depth range and penetration
Limitations	Lower resolution Slower imaging speed Lower SNR Mechanical scanning Lower image quality	Sensitivity roll-off with depth Higher cost Complex technology	Lower axial resolution compared to SD-OCT Higher cost Worse SNR and motion artifacts compared to SD-OCT Limited availability and normative databases	High complexity Very high cost Limited availability and commercialization
Challenges	Mechanical scanning limits speed and efficiency Lower image quality affects diagnostic accuracy	Depth range limitations Higher cost and complexity Requires sophisticated equipment and maintenance	Higher cost and limited availability Lack of normative databases Challenges in clinical integration	High cost and complexity Difficulties in integrating into clinical settings Requires specialized training and maintenance

**Table 3 bioengineering-12-00065-t003:** Summary of applications of artificial intelligence in OCT.

Topic	Details
Artificial Intelligence (AI) in OCT	Not inherently part of OCT systems
	Integration improves signal quality and diagnostic outcomes
	Enhances image quality, disease detection, and processing
	Guides clinical decision making and improves patient management
Deep Learning (DL)	Used for disease differentiation, feature segmentation, and image quality assessment
	Enhances diagnostic accuracy and automates feature extraction
	Contributes to early disease detection and management
Transfer Learning	Leverages pre-trained models on large datasets
	Fine-tunes models with OCT-specific data
	Reduces computational resources and time
	Overcomes challenges regarding limited labeled OCT data
Convolutional Neural Networks (CNNs)	Designed for processing grid-like data such as images
	Automatically learns spatial hierarchies of features
	Consists of convolutional, pooling, and fully connected layers
	High accuracy in image recognition and classification tasks
Support Vector Machine (SVM)	Used for classification and regression tasks
	Differentiates between normal and diseased retinal structures
	Handles both linear and non-linear data using kernel functions
	High accuracy, robustness to overfitting, and handles high-dimensional data

**Table 4 bioengineering-12-00065-t004:** General comparison between the presented methods.

Paper	Method Used	Applications	Accuracy and Resolution	Data Type	Advantages	Disadvantages	Signal Processing by	Wavelength
[87]	µOCT	Imaging prostate specimens	Axial: <1 µm	2D and 3D images	High-resolution imaging	Limited depth penetration (300–500 µm)	--	650 to 950 nm
[88]	PS-OCT	Quantitative T2 MRI mapping for osteoarthritis	Axial: ~7.2 µm, Lateral: ~19.2 µm	3D volumetric data	Fast imaging speed (3D volume in 5 s)	Requires advanced image analysis tools	--	1060 nm
[89]	OCTA	Evaluation of diabetic retinopathy	High resolution (exact values not specified)	Imaging data	Useful for identifying non-perfusion areas	May miss subtle changes	--	850 nm
[92]	Handheld OCT	Imaging of muscle and lumbar fascia	Axial: =15 µm	2D images	Portable and easy to use	Limited to larger needles (e.g., 17 or 18 G)	FPGA	1310 nm
[91]	SSOCT	Establishing peritoneal access	Axial: =15 µm	2D images	Addresses unmet needs in surgical access	Requires development of new tools	FPGA	1310 nm

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
