# Peer review of "A Survey on Optical Coherence Tomography—Technology and Application"

_bioengineering, 2025, doi:10.3390/bioengineering12010065_

Round 1
Reviewer 1 Report
Comments and Suggestions for Authors
The article can be accepted after the minor corrections
1. The paper does a good job summarizing research but lacks critical analysis. Providing a comparative evaluation of different methods, their advantages, limitations, and challenges, would add depth
2. Consider adding more visual aids, such as tables summarizing key studies, charts showing technology evolution, or diagrams explaining OCT principles.
3. Revise for grammatical errors and improve sentence flow to enhance readability.
4. As any researcher knows, if important and useful references are provided in an article, it will enable readers to make good use of it and improve their knowledge and awareness of the research field in question. For this reason, it is strongly recommended to add the following references to the reference section and learn from the way these articles are written to improve the writing quality of your article: doi: 10.37188/lam.2023.036; doi: 10.29026/oea.2024.230212; doi: 10.1109/TMI.2024.3425533; https://doi.org/10.1016/j.saa.2022.122000; doi: 10.3788/COL202018.051701; https://doi.org/10.1016/j.pacs.2023.100569; https://doi.org/10.1002/cjoc.202200406; doi: 10.2147/IJN.S466042; https://doi.org/10.1186/s43074-024-00139-2; doi: https://doi.org/10.1364/PRJ.521056
Comments on the Quality of English LanguageThroughout the manuscript article needs grammar correction
Author Response
- The paper does a good job summarizing research but lacks critical analysis. Providing a comparative evaluation of different methods, their advantages, limitations, and challenges, would add depth
- Thank you for your comment. Table No. 2 has been added to compare the types of OCT technologies.
- Consider adding more visual aids, such as tables summarizing key studies, charts showing technology evolution, or diagrams explaining OCT principles.
- Thank you for your comment. Table No. 1 has been added to show technology evolution
- Revise for grammatical errors and improve sentence flow to enhance readability.
- Thank you for your comment. We have made every effort to address the grammatical issues.
- As any researcher knows, if important and useful references are provided in an article, it will enable readers to make good use of it and improve their knowledge and awareness of the research field in question. For this reason, it is strongly recommended to add the following references to the reference section and learn from the way these articles are written to improve the writing quality of your article: doi: 10.37188/lam.2023.036; doi: 10.29026/oea.2024.230212; doi: 10.1109/TMI.2024.3425533; https://doi.org/10.1016/j.saa.2022.122000; doi: 10.3788/COL202018.051701; https://doi.org/10.1016/j.pacs.2023.100569; https://doi.org/10.1002/cjoc.202200406; doi: 10.2147/IJN.S466042; https://doi.org/10.1186/s43074-024-00139-2; doi: https://doi.org/10.1364/PRJ.521056
- Thank you for the valuable references you suggested. Due to the length limitation of the paper, we could not include a proper discussion of all the references you suggested, and therefore we added to the revised paper those references that are more relevant to the discussion. Our apologies that we could not add all the references suggested.
Reviewer 2 Report
Comments and Suggestions for Authors
Accept with minor revisions.
The revisions are truly minor. May the authors check to see whether the founder of OCT, David Huang, MD PhD is referenced. Also, the authors need to check formatting.
The paper is a real contribution to bioengineering and is ready for publication after these minor revisions.
Author Response
- The revisions are truly minor. May the authors check to see whether the founder of OCT, David Huang, MD PhD is referenced. Also, the authors need to check formatting.
- Thank you for your comment. We have reviewed the formatting and made minor corrections. For instance, some adjustments have been made to the table formatting.
- The paper is a real contribution to bioengineering and is ready for publication after these minor revisions.
- Thank you for your comment.
Reviewer 3 Report
Comments and Suggestions for Authors
The manuscript titled “A Survey on Optical Coherence Tomography – technology and application” is well written. However, there are certain areas which require some clarification. Please find below my comments and suggestion:
1. There are previously reported articles on OCT. Please mention the novelty of this manuscript in the introduction section.
2. The history section is quite elaborated. It would be good if some schematic is added to showcase the timeline rather than writing elaborated text.
3. Please improve the quality of Fig. 1. There is no text explaining this figure. Please add th explanation.
4. Most of the figures used in the manuscript are not explained in main text. It should be explained in main text.
5. The permission for reproducing the figures from published work must be taken and mentioned in the main text.
6. Section 7 can be enriched with more information. This section can be segregated in different subsections for different imaging application such as ocular imaging, plaque imaging etc rather than providing the information in one section.
7. The conclusion section should be improvised significantly. The bullet points in conclusion should be removed and it can be written in con sized manner.
8. The title of the paper and context is not in sync. The introduction states its is “comprehensive overview”. But the manuscript lack in in depth discussion of OCT technique.
9. A table should be added in section 6
10. More information should be added in “software and signal processing” section as it provide very limited information.
11. The Grammer and typo errors must be addressed throughout the manuscript.
Author Response
The manuscript titled “A Survey on Optical Coherence Tomography – technology and application” is well written. However, there are certain areas which require some clarification. Please find below my comments and suggestion:
- There are previously reported articles on OCT. Please mention the novelty of this manuscript in the introduction section.
- Thank you for your comment. Although this explanation was initially provided at the end of the introduction, in response to your recommendation, it has been presented in a more detailed and explicit manner.
- The history section is quite elaborated. It would be good if some schematic is added to showcase the timeline rather than writing elaborated text.
- Thank you for your comment. Table No. 1 has been added to showcase the timeline.
- Please improve the quality of Fig. 1. There is no text explaining this figure. Please add the explanation.
- Thank you for your comment. We have improved the quality of Fig. 1 and added a detailed explanation to ensure clarity and understanding.
- Most of the figures used in the manuscript are not explained in main text. It should be explained in main text.
- Thank you for your comment. References to the figures have been added to the text.
- The permission for reproducing the figures from published work must be taken and mentioned in the main text.
- Thank you for your comment. Although the sources are directly referenced in the images, and I believe that a direct reference should be sufficient, we can obtain permission if necessary.
- Section 7 can be enriched with more information. This section can be segregated in different subsections for different imaging application such as ocular imaging, plaque imaging etc rather than providing the information in one section.
- Thank you for your comment. I completely agree with you regarding the need for more detailed sectioning, elaboration, and explanations. We will follow your indication, within the limitations imposed by the maximun paper length, by restructuring the section in paragraphs to give more emphasis to the different applications.
- The conclusion section should be improvised significantly. The bullet points in conclusion should be removed and it can be written in con sized manner.
- Thank you for your valuable comment. The bullet points in the conclusion section have been removed. However, given the extensive discussions presented in the paper, shortening the conclusion section might compromise the comprehensiveness of the conclusions.
- The title of the paper and context is not in sync. The introduction states its is “comprehensive overview”. But the manuscript lack in in depth discussion of OCT technique.
- Thank you for your comment. The comprehensiveness of the paper refers to its inclusion of all topics from the fields of optical physics, hardware, software, artificial intelligence, and the applications of OCT. When discussing all these areas, it is not possible to deeply examine each field in a single paper.
- A table should be added in section 6
- Thank you for your comment. Table 3 has been added to the text.
- More information should be added in “software and signal processing” section as it provide very limited information.
- Thank you for your comment. Yes, you are absolutely right, and it needs further explanation. More information has been added to the text.
- The Grammer and typo errors must be addressed throughout the manuscript.
- Thank you for your comment. We have made every effort to address the grammatical and typo errors issues.
Round 2
Reviewer 3 Report
Comments and Suggestions for Authors
The authors have revised the manuscript extensively. Now the manuscript can be accepted.